# RT-qPCR Detection of SARS-CoV-2: No Need for a Dedicated Reverse Transcription Step

**DOI:** 10.3390/ijms23031303

**Published:** 2022-01-24

**Authors:** Stephen A. Bustin, Gregory L. Shipley, Sara Kirvell, Reinhold Mueller, Tania Nolan

**Affiliations:** 1Medical Technology Research Centre, Faculty of Health, Education, Medicine and Social Care, Anglia Ruskin University Chelmsford, Chelmsford CM1 1SQ, UK; sara.kirvell1@aru.ac.uk (S.K.); tanianolan@btinternet.com (T.N.); 2Shipley Consulting, Vancouver, WA 98682, USA; gshipley14@me.com; 3RM Consulting, San Diego, CA 92131, USA; reinholdmueller7@gmail.com

**Keywords:** reverse transcription, RNA, molecular diagnostics, RT-qPCR, SARS-CoV-2, COVID-19

## Abstract

Reverse transcription of RNA coupled to amplification of the resulting cDNA by the polymerase chain reaction (RT-PCR) is one of the principal molecular technologies in use today, with applications across all areas of science and medicine. In its real-time, fluorescence-based usage (RT-qPCR), it has long been a core technology driving the accurate, rapid and sensitive laboratory diagnosis of infectious diseases. However, RT-qPCR protocols have changed little over the past 30 years, with the RT step constituting a significant percentage of the time taken to complete a typical RT-qPCR assay. When applied to research investigations, reverse transcription has been evaluated by criteria such as maximum yield, length of transcription, fidelity, and faithful representation of an RNA pool. Crucially, however, these are of less relevance in a diagnostic RT-PCR test, where speed and sensitivity are the prime RT imperatives, with specificity contributed by the PCR component. We propose a paradigm shift that omits the requirement for a separate high-temperature RT step at the beginning of an RT-qPCR assay. This is achieved by means of an innovative protocol that incorporates suitable reagents with a revised primer and amplicon design and we demonstrate a proof of principle that incorporates the RT step as part of the PCR assay setup at room temperature. Use of this modification as part of a diagnostic assay will of course require additional characterisation, validation and optimisation of the PCR step. Combining this revision with our previous development of fast qPCR protocols allows completion of a 40 cycle RT-qPCR run on a suitable commercial instrument in approximately 15 min. Even faster times, in combination with extreme PCR procedures, can be achieved.

## 1. Introduction

The discovery of RNA-dependent DNA polymerases [1,2] ranks as one of the most consequential findings in modern biology. The ability of these enzymes, also known as reverse transcriptases, to convert RNA into DNA has been a central driver of progress in modern biotechnology. Combined with the subsequent discovery of *Taq* polymerase [3] and the theory [4] and practical realisation of the polymerase chain reaction (PCR) [5,6], reverse transcription (RT)-PCR was quickly adopted as a powerful method for the analysis of RNA [7,8]. Whilst early protocols included 60 min incubation times for the RT step [9], most contemporary protocols tend to be more rapid at between 15 [10] and 30 min [11], although markedly longer RT polymerisation times continue to be used [12]. This is generally not a problem for research applications but becomes an issue when RT-PCR is used in the context of rapid diagnostic testing. Examples are intraoperative testing for micrometastases [13], detection of viral pathogens in humans [14] or corresponding applications in veterinary medicine [15]. The need for speed has become apparent most recently as RT-qPCR has enabled early mass testing of individuals for severe acute respiratory syndrome coronavirus 2 (SARS-CoV-2), the aetiological agent of coronavirus disease 2019 (COVID-19). Sustained testing remains an indispensable tool for helping to limit infectious spread between individuals and communities, especially given the threat posed by asymptomatic infected people.

Having previously demonstrated the feasibility of using a short one-to-five min RT step prior to qPCR assays [14,16], we set out to establish the minimum RT incubation time and temperature required to complete transcription of a target RNA. We chose to focus on the RNA genome of SARS-CoV-2 as this pathogen is currently at the centre of efforts to develop point-of-care (POC) devices for rapid diagnostic testing. We designed a series of short amplicons as well as primers with standard and non-standard hybridisation characteristics and used several RT enzymes to develop an RT-qPCR protocol that no longer required a dedicated RT step. Instead, against prevailing orthodoxy [17], reactions were set up at room temperature with RT reaction times limited to the time it takes to prepare the reaction mixes, pipette them, seal, spin the PCR plates and place them inside the qPCR instrument. The procedure was implemented with several commercial master mixes and was shown to work with both 1-step and 2-step RT-qPCR protocols. We anticipate that this innovation will provide a ground-breaking new standard for future RT-qPCR protocols and result in reduced assay times and simpler POC instrumentation.

## 2. Results

### 2.1. 2-Step RT-qPCR 

#### 2.1.1. Effect of Temperature

SARS-CoV-2 RNA was reverse transcribed with SS4 using four RT conditions followed by the recommended 5 min RT inactivation at 85 °C: (i) the prescribed 10 min at 25 °C followed by 10 min at 50 °C, which served as the positive control, (ii) 5 min at 50 °C without the initial incubation at 25 °C, (iii) 5 min at 30 °C and (iv) 5 min at room temperature (21 °C). Aliquots of the resulting cDNA preparations were amplified with primers targeting the viral E-gene (assay E1, Table 1) and detected using a hydrolysis probe on a BioRad CFX instrument (Figure 1A). 

There was little difference in Cq among the four RT conditions tested (Figure 1B), with average ∆Cqs of −0.27 (−0.39 to 0.15), −0.04 (−0.12 to 0.03) and 0.18 (0.08 to 0.27) for protocols (ii), (iii) and (iv) and control protocol (i). Reactions (ii) and (iv) were repeated using SYBR Green detection and again, there was no difference in Cq. Melt curve analysis showed single peaks that were identical at both RT temperatures and differed from the NTC (Figure 1C). RNA stability is not affected by leaving it at room temperature for up to 24 h, as demonstrated in Figure 1D.

A second assay, N1 targeting the Nsp10 gene (Table 1), recorded similar results. The ∆Cq values between the 21 °C (iv) and recommended RT protocol (i) were slightly higher at 0.31 (range 0.3 to 0.32). (Appendix A). When the 50 and 21 °C reactions were repeated with SYBR Green as the reporter, there was a 1 Cq difference between the two RT temperatures (Appendix A). Melt curve analysis showed identical single peaks at both RT temperatures, which were distinct from the NTC (Appendix A).

The experiment was repeated with both assays and four conditions, except that the qPCR step used probe-based detection and assayed using the PCRMax Eco instrument. Results for E1 were similar (Appendix A), with the ∆Cqs between the standard RT protocol (i) and the 50 °C (ii), 30 °C (iii) and 21 °C (iv) modifications being 0.07 (−0.36 to 0.45), −0.04 (−0.49 to 0.37) and 0.27 (−0.24 to 0.64) (Appendix A), respectively. The ∆Cqs for the N1 assay were slightly wider at 0.34 (−0.12 to 1.17), 0.39 (0.03 to 0.97) and 0.71 (0.14 to 1.37) (Appendix A). Cq values are listed in the Appendix A.

The activity of SS4 at a range of low RT temperatures was investigated by performing an eight-step RT gradient ranging from 30 to 50 °C and amplifying the resulting cDNA samples targeting the E-gene (assays E1, E2 and E3, Table 1) as well as Nsp10 (assays N1, N2 and N3, Table 1). A second RNA sample was amplified with the E1 assay using the same RT gradient and amplification conditions. Results confirmed the ability of the SS4 enzyme to transcribe RNA efficiently at all temperatures tested, with the independent replicate for E1 giving essentially the same result (Figure 2A). The experiment was repeated with SYBR Green used as the reporter. The results were effectively the same (Figure 2B). Melt curves for the six assays showed single peaks at all RT temperatures. High background fluorescence was due to the presence of the probe (Figure 2C). The Cq values are listed in the Appendix A.

To further investigate the relationship between each assay and RT at temperatures below 30 °C, a second RTase, US2, was used to reverse transcribe RNA alongside SS4 for 5 min at either 50 °C (ii) or 21 °C (iv) The resulting cDNAs were amplified using the six assays targeting the E and N genes. The Cq patterns obtained with the two RTases were similar, although SS4 enzyme (Figure 3A) recorded lower Cqs than US2 (Figure 3B). Moreover, the ∆Cqs between the 50 and 21 °C RT protocols were smaller with SS4, indicating that SS4 was more efficient at reverse transcribing the RNA at the lower temperature (Figure 3C). RT variability was assessed by carrying out three and nine independent RT reactions, respectively, at 50 or at 21 °C with all six assays using SS4 (Figure 3D). The average ∆Cqs and 95% confidence intervals were 0.28 (0.09–0.48), 0.54 (0.36–0.71), 0.54 (0.37–0.71), 0.52 (0.29–0.74), 0.83 (0.64–1.0) and 0.47 (0.38–0.56) for E21, E2, E3, N1, N2 and N3, respectively.

A repeat experiment with two further RNA sample gave similar results, with SS4 recording lower ∆Cq values than US2 (Appendix A). Analysis of the melt curves from the experiment carried out with SYBR Green as the reporter showed identical single peaks for the E2 and E3 PCR amplicons (Appendix A) as well as the N2 and N3 amplicons at both temperatures (Appendix A), as with E1 and N1 previously. The Cq values are listed in the Appendix A.

Given that assay E1 gave the most consistent results, its reproducibility was analysed in more detail by carrying out additional independent RT-qPCR assays with RT steps at 50 or 21 °C. First, the same RNA sample was transcribed by SS4 in four replicate reactions for 5 min at 50 °C and in nine replicate reactions at 21 °C and amplified on both the CFX and Eco instruments. Results between instruments were comparable (Appendix A), showing that assay E1 is efficiently transcribed at 21 °C by SS4 independently of which instrument was used. Two further RNA samples were transcribed by SS4 for 5 min at 50 or at 21 °C and amplified on the CFX, giving similar results (Appendix A). The ∆Cq values for the RT reactions ranged from −0.81 to 2.13 on the CFX and −1.41 to 1.42 on the Eco. The Cq values are listed in the Appendix A.

#### 2.1.2. Priming on Ice

The ability of RTases to transcribe at even lower temperatures was tested by carrying out the following reactions in duplicate: RNA samples in RT reaction buffer were split in two, one being reverse transcribed with SS4 or US2 and random hexamers at 50 °C, the other set left on ice for 5 min, followed by amplification and detection of assays N1, N2 and N3. To our surprise, both RT enzymes showed significant activity on ice, with average ∆Cqs for SS4 and US2 of 2.8 (2.25–3.38) and 3.57 (1.6–3.57), respectively; a repeat experiment generated equivalent results (Appendix A). The experiment was repeated with ten independent RT reactions carried out with US2 for 5 min at 50 °C, 21 °C or on ice and the E1 assay was amplified on the CFX. The RT reactions carried out at 21 °C showed spread of ∆Cqs similar to the ones observed for SS4, with an average ∆Cq of 0.45 (range −0.77 to 2.06). RT efficiency was lower and a little more variable when the RT step was carried out on ice, with an average ∆Cq of 3.3 (range 1.39 to 4.87). A third experiment was carried out using SS4 to reverse transcribe RNA at 50, 30, 21 and 0 °C for 5 min. The cDNA was amplified with the E1 assay and the results confirm that SS4 is more efficient than US2 at reverse transcribing at 21 °C and fairly efficient at 4 °C (Appendix A). Finally, a different RNA sample was reverse transcribed with SS4 and random hexamers at 50, 21 and 0 °C, followed by qPCR amplification targeting all six assays and detection of PCR amplicons by SYBR Green. Assays E1, E2 and E3 assays showed single peaks for each assay, with the same Tms for samples reverse transcribed at all three temperatures (Appendix A). Assay N1 also recorded a single peak at the three temperatures, but assays N2 and N3 had an additional, smaller peak at 4 °C (Appendix A). All Cq values are listed in the Appendix A.

A time course of RT priming with random hexamers was carried out comparing results from four RT replicates incubated on a thermal cycler for 5 min at 50 °C to those obtained by carrying out the RT step at 30, 21 or 0 °C for 2.5, 5 or 10 min. The resulting cDNA was amplified by a single qPCR using assays E1 or N1. Comparing the Cqs to the 50 °C control confirms that the option of carrying out the RT step as part of the reaction setup is feasible (Figure 4A,B). Amplification plots for both targets show similar amplification efficiencies of the cDNAs (Figure 4C,D) and the melt curves show single PCR amplicons for both (Figure 4E). All Cq values are listed in the Appendix A.

#### 2.1.3. Priming Effects

The effect of combining random hexamers and assay-specific primers on reverse transcription was analysed by carrying out the RT reactions at 50 or 21 °C compared with random hexamers alone or a combination of hexamers and assay-specific reverse primers using SS4 or US2. On this occasion the reactions primed with SS4 at 21 °C with only random hexamers resulted in lower Cqs for all six assays compared with those primed at 50 °C. Addition of specific primers enhanced the efficiency of the RT reaction at 50 °C, with lower Cqs recorded for all six assays (Figure 5A). In contrast, there was no marked effect at 21 °C. The results obtained with US2 were similar, although the effects of adding random hexamers were less pronounced at 50 °C and there was less uniformity at 21 °C (Figure 5B). The Cq values are listed in the Appendix A.

### 2.2. 1-Step RT-qPCR Cqs

The 1-step RT-qPCR reactions with the E1 assay were performed with Takara’s PrimeScript 3, PCRBio’s OneStep Go, Sigma’s RT and GSD’s NovaPrime master mixes using a BioRad CFX instrument. Three RT reaction conditions were used: (a) 10 min at 50 °C, (b) 1 min at 50 °C, or (c) 5 min at 21 °C, followed by our standard qPCR amplification protocol (40 cycles at 95 °C and at 60 °C, both for 1 s). The results show that RT efficiency was similar under all three conditions, with PrimeScript and NovaPrime the most efficient at 21 °C (Figure 6A).

The experiment was repeated, with two replicate RT reactions for each of the 1-step master mixes carried out for 5 min at 50 or 21 °C. Reactions and amplifications were carried out on the PCRMax Eco instrument. The ∆Cqs recorded in these experiments were comparable to those recorded earlier, with the ∆Cq spread confirming that PrimeScript 3 gave the least variable results (Figure 6B). A third qPCR instrument, the BMS Mic, was used to repeat one of the RT-qPCR experiments shown in Figure 6B. This time only PrimeScript was used. The 5 min RT reactions were carried out separately from the amplification in a 50 °C water bath or left at room temperature. The experiments were carried out on consecutive days, and ten qPCR replicates were analysed. The results were concordant with those obtained on the CFX and Eco qPCR instruments (Figure 6C). All Cqs are listed in the Appendix A.

The ability of seven 1-step master mixes to reverse transcribe RNA at a range of lower temperatures was investigated by performing the RT step with target-specific hexamers on an eight-step gradient ranging from 30 to 50 °C and amplifying the resulting cDNA samples with the E-gene assay. All master mixes except NEB’s Luna recorded similar Cqs across the 20 °C gradient, showing that the efficiency of the RT step for this assay was no different at 30 °C than it was at 50 °C (Figure 7A). The experiment was repeated with PrimeScript, OneStep Go and Sigma master mixes utilizing assays targeting the E1 and N1 genes. Results show both RT- and assay-dependent differences (Figure 7B). The reproducibility of the results was assessed by repeating the RT-qPCR reactions for the best-performing assay, E1, simultaneously using the gradient function at 50 and 30 °C as well as separately at 21 °C with PrimeScript 3 (Figure 7C). Melt curve analysis of confirmed the amplification of a single PCR amplicon with the expected Tm at RT temperatures of 50 and 30 °C (Figure 7D) as well as 21 °C (Figure 7E). The Cq values are listed in the Appendix A.

The RT component of PrimeScript 3 has the highest activity at 4 °C, amongst the 1-step kits tested as determined by analysing the expression of assays E1 and N1. Reagents were assembled on ice and dispensed into cold qPCR plates, which were left on ice, inadvertently, for 15 min. Results from five individual RT-qPCR reactions for each assay were compared to those obtained following a standard 5 min RT step at 50 °C. Surprisingly, the E1 and N1 were as efficiently transcribed at 0 °C as at 50 °C (Figure 8A). To determine whether a longer incubation period at 0 °C enhanced RT efficiency, a time course of RT reactions left on ice for 15, 10 or 5 min was tested, followed by amplification of the E1 target. Cqs were compared to a standard RT-qPCR assay carried out at 50 °C (Figure 8B). The results showed a time-dependence, with average ∆Cq values of 1.0 ± 0.4, 1.9 ± 0.5 and 3.1 ± 0.4, respectively, for the 15, 10 and 5 min RT times. The time course was repeated for RT reactions kept at 21 °C, with average ∆Cq values of 0.3 ± 0.2, 0.6 ± 0.2 and 1.0 ± 0.3, respectively, for the 15, 10 and 5 min RT incubations relative to the control RT assay (Figure 8C). The Cq values are listed in the Appendix A.

The results obtained with random hexamer priming using the 2-step RT protocol prompted us to explore the RT priming issue in more detail with 1-step RT-qPCR reactions, even though these are generally carried out using specific primers only. Two sets of reactions were carried out using PrimeScript 3, with a 5 min RT step at either 50 or 30 °C using one plate and the CFX gradient function: the first set used assay-specific primers only at 500 nM, the second set supplemented the assay-specific primers with 100 nM final concentration of random hexamers. Melt analysis of the resulting PCR amplicons were detected with SYBR Green to demonstrate assay specificity. The random hexamers had only a small effect on the sensitivity of the assays, apart from N1 at 50 °C (Figure 9A). Melt curve analysis confirmed the specificity of the detected PCR amplicons with similar Tms at both RT temperatures (Figure 9B). The experiment was repeated, this time with separate RT steps at 50 and at 30 °C. Reactions were amplified on the Eco instrument. Results for both temperatures were similar (Figure 9C), although there was a possible trend for the three assays targeting the E-gene recording lower Cqs with hexamers at 30 °C. However, when the reactions with the three assays targeting the E-gene were repeated on the CFX at both temperatures, this trend was not confirmed. The N1 gene again recorded lower Cqs in the presence of hexamers at 50 °C. Melt curve analysis confirmed the specificity of the PCR amplicons (Figure 9D). All Cq values are shown in the Appendix A.

## 3. Discussion

We have developed an innovative RT protocol that satisfies two fundamental requirements for a diagnostic assay: (i) the obligation to maintain both specificity and sensitivity and (ii) the capacity to transcribe templates with significant secondary structure. Our rapid, low-temperature approach challenges the traditional expectations for the performance of RT reactions. These have been shaped by the role of RNA-dependent DNA polymerases in cDNA synthesis, cloning, expression analysis and sequencing. These functions require processive enzymes that have a minimal mutation rate, can reliably transcribe rare transcripts and generate large amounts of cDNA. This has resulted in the development of a range of different reverse transcriptase enzymes (RTases) with diverse performance characteristics [18] that enable each one to meet the needs of distinct applications [19]. For both research and diagnostic applications, RT reactions usually last between 15 and 30 min [10,11] using as high a temperature as possible [20]. This creates a substantial drawback for diagnostic tests aimed at rapid viral pathogen detection, especially when carried out in a point-of-care setting. We have previously shown that RT reactions can be completed in 1 min [16]. Following on from this initial observation, we now report by choosing the right RTase, primers and short PCR amplicon, the RT step can be carried out efficiently and with comparable sensitivity at remarkably low temperatures, even with a highly structured RNA template. This can be done without the historically requisite separate RT step. 

Our finding promises a paradigm shift for RT-qPCR-based diagnostic applications. The proposed new protocol comprises the dispensing of reagents, addition of template and pipetting into qPCR plates at room temperature. This setup time is sufficient to allow the RTase to transcribe past the binding site of the forward PCR primer. The approach works for both 1-step and 2-step protocols as well as with samples that contain lots or little viral RNA, but is dependent on choice of RTase, characteristics of primers and length of the PCR amplicon. We have not assessed the influence of different extraction protocols, which could affect RNA integrity.

These findings are rationalised by a recent report that shows RTases retain activity at 21 °C, incorporating approximately 10 nucleotides per second [21], giving the enzymes plenty of time to transcribe through the forward priming binding sites of short amplicons, which in this report are between 56 and 77 nucleotides in length. Unfortunately, the authors did not measure RTase speed below 21 °C but, interestingly, the 10 nucleotides per second rate remained constant between 21 and 33 °C. Our results certainly demonstrate that, to our own surprise. We have demonstrated sufficient activity remains at 0 °C to transcribe short amplicons fairly efficiently in 5–10 min. Furthermore, whilst wild-type RTases are not highly processive, falling off their template after incorporating 20–30 nucleotides, todays engineered versions are vastly more processive, with the best-performing variants remaining on templates after 1500 nucleotides [22]. Hence, this protocol addresses two key requirements for a diagnostic assay, speed and sensitivity. Importantly, fidelity is not a problem for this specific application, since avian myeloblastosis or murine leukaemia virus RTases have average error rates of 1 per 17,000 and 1 per 30,000 bases, respectively [23]. Whilst short assay times and absence of a dedicated RT step may not constitute major advantages, or indeed be of prime importance for time-tolerant research applications, these modifications offer the potential for designing simplified point-of-care devices with the same assay conditions for RNA or DNA targets. 

It has long been established that carrying out reverse transcriptions at higher temperatures increases the specificity and sensitivity of the reaction and enhances the processivity of the RTase [20]. Hence, there have been extensive efforts to improve the thermal stability and performance of RTases [24]. This work has resulted in a range of mutated or engineered RTases [22,25,26,27] able to synthesise cDNA at temperatures well above the original 37–42 °C [28,29]. Other approaches make use of the ability of *Tth* polymerase to act both as an RNA- and a DNA-dependent DNA polymerase [30], combine *Tth* polymerase with a thermostable RTase [31] or even use *Taq* polymerase with modified buffers in a one enzyme protocol [32]. However, the latter approach resulted in lower RT efficiencies than achieved by an authentic RT. Furthermore, buffers may need to be optimised for different templates. One intriguing alternative method bypasses the RT step altogether and gives a sample-to-signal time of under 10 min [33], although its requirement for a hybridisation and nicking step relies on the identification of optimal nucleotide sequences in the target genome, which may not always be present.

An additional concern driving ever-higher RT reaction temperatures is the assumption that RNA secondary structure issues are reduced at higher temperatures, making primer choice and location an essential component of any assay design using specific primers for the RT reaction [34,35,36]. This is thought to be of particular concern when targeting viral genomes, where significant secondary structures may block primer binding and impede nascent strand synthesis on the RNA template [17,37]. There is a wealth of information with regard to primer requirements for PCR [38,39]. Furthermore, priming bias at the primer-template junction during the initiation of polymerisation is a known issue [40]. In contrast, little is known about the equivalent effects on RNA-dependent DNA polymerases.

We addressed these issues by designing short amplicons. Thus, the RTase only needed to transcribe a short stretch of RNA for E1 (51 nucleotides from the 3′-end of the primer), E2 (55 nucleotides), E3 (37 nucleotides), N1 (51 nucleotides), N2 (42 nucleotides) and N3 (38 nucleotides), respectively. The primers were of variable design: one ended in CG, four in AG and one in AT. They also differed in their melting temperatures, which ranged from 65.2 to 56.8 °C. Figure 1 and Appendix A show a comparison of the results from 2-step RT-qPCR assays for assays E1 and N1 carried out with SS4 using variable RT times and temperatures. Both PCR amplicons were the same length, but the N1 primers had lower Tms and the N1 reverse primer ended in AG, compared to the E1 reverse primer that ended in CG. The results show there was little difference in the ability of SS4 to prime transcription from either primer and that priming efficiency was remarkably temperature-tolerant. The single melt curves obtained for each amplicon also demonstrated that the subsequent amplification products were assay specific. Our experiments also addressed the commonly held view that purified RNA is unstable. We show this is clearly not the case even after 5 days at room temperature, at least for the short amplicons being targeted (Figure 1D and Appendix A). Furthermore, there were no differences attributable to the qPCR instrument (Appendix A), something that might have been expected since the BioRad CFX initiates the RT step much more slowly than the Eco instrument, providing more time for the RT to be active on the samples incubated at room temperature. The CFX also has a much slower ramping rate, which might be anticipated to provide a longer, optimal temperature window for the RT reaction.

The influence of RNA secondary structure on RT priming efficiency is an interesting, yet under-explored component of RT-qPCR sensitivity. Using the data from a recent publication that modelled the secondary structure of the SARS-CoV-2 genome [41], we surmised that the reverse primers from all six assays hybridise to regions of the genome projected to contain extensive secondary structures. Hence, we did not expect the minimal effect on priming efficiency that we observed. We used additional modelling with separate secondary structure prediction programmes [42,43], which both confirmed that the reverse primers for the E-gene (Appendix A [42] and Appendix A [43]) and the N-gene (Appendix A [42] and Appendix A [43]) targeted areas of extensive secondary structure (Appendix A). Both models also agreed that whereas the 3′-ends of primers for the E1, N1 and N2 assays are within a predicted secondary structure, the two oligonucleotides priming the E2,3 and N3 assays and in a predicted loop. In practice, this has little effect; indeed, assay E1 results in the most reliable, efficient and sensitive assay of the set. It is of course important to bear in mind that these models, even though they agree, may not reflect the structures found in vivo or under reaction conditions.

Carrying out experiments at room temperature is not ideal, as the temperature can fluctuate day-to-day in one lab and certainly will differ between labs. Hence, we repeated the experiments with six assays on an RT gradient using the BioRad CFX. These experiments were limited only by the inability of the instrument to cool to below 30 °C. The results revealed a remarkable consistency in the Cq values recorded at the eight temperatures with each of the six assays, showing that primer Tm or 3′-sequences(s) were of little importance (Figure 2A). There was also no obvious association between amplicon length and RT efficiency, although even the longest amplicon (E2) is short at 77 bp by qPCR standards. When the experiment was repeated with SYBR Green I as the main reporter, results were similar, and although there was a trend toward lower Cqs at higher RT temperature for some of the assays, this was neither consistent nor significant (Figure 2B). Melt curve analysis confirmed the previous observation that single amplification products were generated at all temperatures. The high background fluorescence at higher temperatures was due to the presence of probe as well as SYBR Green in the reactions (Figure 2C).

Figure 3 and Appendix A show that the results obtained with SS4 are not unique to that RTase. A second RTase, US2, was also able to transcribe RNA into cDNA, but results were subtly different. US2 was consistently less efficient than SS4, as shown by a comparison of Cqs recorded for the six assays (Figure 3 and Appendix A). At the same time, the ∆Cqs between the standard 50 °C RT reaction and that carried out at 21 °C were larger for most of the assays, with the lower efficiency seen at 21 °C. Differences between the two enzymes were particularly obvious for N1 to N3 (Figure 3 and Appendix A). As with E1 and N1, melt curve analyses for assays E2, E3, N2 and N3 revealed identical single peaks for the E2 and E3 PCR amplicons reverse transcribed at 50 and 21 °C (Appendix A) as well as the N2 and N3 amplicons (Appendix A). More detailed analysis confirmed that the results were reproducible and instrument- as well as RT-independent for the most consistent of the assays, E1. A direct comparison of the same samples run on the CFX and Eco instruments resulted in average ∆Cq values of 0.23 ± 0.6 and a ∆Cq range of −0.8 to 2.1 on the CFX and −0.21 ± 0.7 and −1.4 to 1.4, respectively, on the Eco (Appendix A). A comparison of two samples run on the CFX resulted in average ∆Cq values of 0.46 ± 0.4 and 0.45 ± 0.2, with a combined ∆Cq range of −0.4 to 1.4 (Appendix A). These data are similar to the ones recorded by the ten independent replicates transcribed with US2, which resulted in an average ∆Cq of 0.45 ± 0.7 and a ∆Cq range of −0.8 to 2.1 (Appendix A).

Given the unexpected activity of RTases at 21 °C, we decided to determine how much activity RTases retained when left on ice compared to RT reactions carried out at 50 °C. Differences in the efficiency of transcription for both SS4 and US2 between 4 and 8-fold were much smaller than anticipated for the three Nsp10 assays initially assessed (Appendix A). When US2 was used in ten independent RT reactions at 0 or 50 °C using the E1 assay, the average ∆Cq was 3.32 ± 1, i.e., 10-fold, with a range between 2 and 32-fold (Appendix A). A repeat reaction with SS4, including an additional 30 °C RT incubation with assay E1, gave similar results at 30 and 21 °C to those obtained earlier but showed an even smaller difference at 0 °C (Appendix A). Melt curve analysis revealed single melt curves with the same Tms for amplicons amplified from 50 or 0 °C RT reactions (Appendix A). These results were reproducible, as demonstrated by experiments targeting E1 and N1 using a time course of RT reactions carried out at 30 °C, 21 °C or left on ice for 2.5, 5 or 10 min. At 30 and 21 °C, RT reactions were complete after 2.5 min and 5 min, respectively, compared with results obtained at 50 °C (Figure 4A,B). The RT reactions left on ice were much less active but showed unanticipated activity. Melt curves for qPCR reactions carried out on all templates confirmed that single, specific PCR amplicons had been generated (Figure 4E).

All RT reactions reported above were initiated by random hexamers, which may help explain why RT efficiency was not hugely different at lower temperatures. Hence, we looked at the effects on RT efficiency of combining hexamers with specific primers. Our expectation was that whilst addition of the specific primers might increase the sensitivity of the assays at 50 °C, it was unlikely to do so at 21 °C. This was indeed the case with the effect of adding specific primers to the 50 °C RT reaction carried out using SS4 resulting in lower Cqs for all six assays (∆Cq = −1.8 ± 0.28 (E1), −1.35 ± 0.22 (E2), −1.14 ± 0.13 (E3), −0.82 ± 0.06 (N1), −0.87 ± 0.15 (N2) and −1.08 ± 0.12 (N3). In contrast, there was no significant effect at 21 °C (∆Cq = −0.16 ± 0.09 (E1), −0.12 ± 0.10 (E2), 0.06 ± 0.14 (E3), −0.14 ± 0.10 (N1), −0.08 ± 0.14 (N2) and 0.05 ± 0.14 (N3) (Figure 5A). US2 gave equivalent results (Figure 5B), with the differences in apparent RT efficiency within the experimental variability expected from RT reactions [18]. By chance, with SS4 this experiment recorded lower Cq values for all six assays at 21 °C when the RT reaction was carried out in the presence of hexamers only. For example, the Cq for the RT reaction carried out at 50 °C with hexamers was 28.54 ± 0.28 compared to 27.05 ± 0.09 for the reaction carried out at 21 °C (Appendix A). Since addition of specific primers increased the sensitivity of the assay transcribed at 50 °C, the Cqs for the RT reactions carried out at 50 °C with hexamers and specific primers were similar, again with assay E1 recording a Cq of 26.74 ± 0.21 at 50 °C compared to 26.88 ± 0.09 for the reaction carried out at 21 °C. On this occasion this was not the case with US2, where only the assays targeting the Nsp10 gene showed this pattern Appendix A). Again, it is unclear whether these results constitute real differences, given the known variability of the RT reaction as well as the levels of variability described in this report, with only the consistency of the increased Cq indicating that this might be a real difference. Certainly, variability associated with carrying out the RT step at lower temperatures could be reduced by carrying out the “21 °C” reactions, which really are an average of fluctuating lab temperatures, at a constant 30 °C, for example by designing a work platform that hold tubes, plates, RNA and reagents at a constant 25°C or 30 °C.

For practical reasons many diagnostic tests are developed as 1-step reactions. It was essential to investigate whether the 2-step results described above translated into viable 1-step reactions. Results obtained with four 1-step master mixes revealed that carrying out the RT at 21 °C resulted in an efficiency comparable to that at 50 °C (Figure 6A). Similar results were obtained when two further replicate reactions were amplified on the Eco (Figure 6B). There were some potential differences between master mixes, but in each case the ∆Cq spread obtained with PrimeScript 3 gave the least variable results. Two additional RT reactions carried out at 21 °C were amplified on the BMS Mic instrument, with equivalent results (Figure 6C). Consequently, we concluded that this was a more general finding than one confined to a single type of RT reaction, since results obtained at the lower temperature RT step were as reproducible as with the 2-step protocol. Again, there appeared to be no need for a dedicated RT step, especially for PrimeScript 3.

An RT temperature gradient ranging from 30 to 50 °C performed with seven 1-step master mixes showed the reaction efficiency held true across a range of temperatures, that there was at least one type of RTase not suited to a low-temperature RT reaction. In addition, we found some assay-dependent distinctions.

Whilst the 1-step results were comparable to the ones obtained earlier with SS4, the most obvious difference was the result recorded with NEB’s Luna, which was inefficient at reverse transcribing at temperatures below approximately 42 °C (Figure 7A). We had anticipated this, since the RT is marketed as a “warm start” enzyme requiring an activation step due to reversible, aptamer-based inhibition of the RT activity. The manufacturers do not recommend incubation at temperatures lower than 50 °C. This enzyme is therefore unsuitable for low-temperature RT, which is of course the rationale behind engineering aptamer inhibition. However, given our results, it does beg the question whether there is a need for low-temperature inhibition. A repeat of the RT gradient with three of the master mixes and the E1 and N1 assays showed apparent master mix and assay-dependent differences (Figure 7B). This was especially apparent with the Sigma master mix, which was less efficient at lower RT temperatures and generated higher ∆Cq values at 30, 31.2, 33.8, 37.6 and 42.5 °C using the N1 assay. A repeat of the E1 assay at 50, 30 and 21 °C using multiple RT replicates confirmed that this assay can be efficiently (Figure 7C) and reliably (Figure 7D,E) reverse transcribed without a dedicated RT step.

The 1-step protocol was also assessed for RT activity at 0 °C using PrimeScript 3, followed by amplification of the E1 and N1 assays. The reactions were left on ice for 15 min and recorded Cqs unexpectedly close to the control 50 °C reactions (Figure 8A). An investigation into the effects of leaving the RT reaction for longer times at either 0 or 21 °C showed that whilst a longer incubation time made some difference at both temperatures, it was surprisingly small (Figure 8B,C), especially at 0 °C considering that priming was dependent on a specific primer binding to a target restrained by extensive secondary structure.

Whilst 1-step RT-qPCR assays are carried out using specific primers for the RT step, the results obtained with the 2-step assays encouraged us to investigate what the effects of adding random hexamers might be and whether they might affect the efficiency of the lower temperature RT step (Figure 9). To be able to study the specificity of the reactions, we used SYBR Green as the reporter, allowing us to analyse the melt curves. We repeated the 1-step RT-qPCR reactions using assay-specific or a combination of specific and random hexamer priming, initiated simultaneously using the CFX gradient function at the two extremes of 50 and 30 °C, followed by separate reactions at 50 and 21 °C. We found the inclusion of random primers made little difference to the Cqs at either RT temperature, with the possible exception of assay N1, which was more sensitive in the presence of hexamers at 50 °C. The melt curves indicated that PCR amplicons were specific, regardless of which priming method was used. The differences in sensitivity between priming at 50 or at 21 °C were in the same range observed with the 2-step protocol but were RTase-dependent.

Finally, low-temperature RT works equally well with samples containing higher or lower copy number targets. The former are loosely defined as samples with Cqs of 20 and below, the latter of 30 and above. This was true for both 1-step and 2-step assays, with examples shown in Appendix A. Furthermore, low copy numbers did not affect the ability to identify mutations, as shown by the detection of the single G to A transition responsible for the E484K mutation in the spike protein (Appendix A). All Cqs are listed in the Appendix A.

In conclusion, we have demonstrated that RT reactions can be carried out successfully and efficiently as part of the reaction setup prior to carrying out a qualitative RT-qPCR test on a PCR instrument. We find this requires the use of a suitable reverse transcriptase in combination with the assay design. We have determined conditions that are signally suitable for POC diagnostic assays, are reproducible and can detect the same target repeatedly with high reproducibility. Obviously, this proof of principle deals only with the RT step and its incorporation into a diagnostic assay will require a characterisation of the limits of detection and linearity of the associated PCR step. This approach could also be used with laboratory-based testing, where the development of a simple workstation to hold tubes and plates at a constant 30 °C would allow also obviate the need for a separate RT step. Whilst it is likely that short amplicons are likely to give the most consistent results, the detailed parameters governing optimal primer characteristics, amplicon size and target location still need to be established. Finally, it would be worthwhile to search for or engineer RTases with optimal activity at low temperatures to achieve maximum sensitivity.

## 4. Materials and Methods

### 4.1. Instruments, Reagents and Analysis

Reactions were carried out on the following instruments: CFX (BioRad, Watford, UK), Eco (ColePalmer, St Neots, UK), Prime Pro (ColePalmer, St Neots, UK) or Mic (BMS, London, UK). The details of all reagents, plasticware and instruments are listed in the Appendix A reagents etc Table Absence of contamination was determined by running no template (NTC) and no RT (NRC) controls. Data were analysed using instrument software, Microsoft Excel for Mac v.16.53 and PRISM for Mac v.9.2.0.

### 4.2. RNA

All assays were carried out using RNA extracted from anonymised patient samples collected four routine diagnostic testing and obtained from Mid and South Essex NHS Foundation Trust Broomfield as previously described [16].

### 4.3. Primer and Probe Design

The human SARS-CoV-2 (NC_045512.2) reference sequence was downloaded to the Beacon Designer 8.21 qPCR assay design software package (Premier Biosoft, San Francisco, CA, USA). Primers and probes targeting the E-gene (E) and Nsp-10 regions (N) of SARS-CoV-2 were designed with manual adjustments aimed at obtaining the shortest possible amplicon sizes amplified by a range of primers with different melting temperatures (Tm). The specificity of primers, probes and amplicons was analysed in silico using Primer-BLAST (https://www.ncbi.nlm.nih.gov/tools/primer-blast/, accessed on 14 November 2021) and BLAST (https://blast.ncbi.nlm.nih.gov/Blast.cgi, accessed on 14 November 2021). All oligonucleotides were synthesised by Sigma-Aldrich (Havershill, UK). Upon receipt, all oligonucleotides were resuspended in sterile RNase-free water at 100 µM and stored in aliquots at −20 °C. Table 1 lists the details of all oligonucleotides and amplicons.

### 4.4. 2-Step RT-qPCR Reactions

Conventional RT reactions were set up on ice using pre-cooled reagents. RNA was reverse transcribed in replicate 5 µL volumes in 0.2 µL thin-walled microfuge tubes using either a complete Superscript IV Vilo (SS4) one-tube master mix or by combining RNA, Ultrascript 2.0 (US2), 2 µM random hexamers and buffer in one tube, at times supplemented with 100 nM specific reverse primers. Following a 5 s spin at 3K RPM in a microfuge, the tubes were transferred to a conventional Thermocycler (G-Storm, Catcombe, UK) with the heated lid set to 112 °C and incubated using the following alternative protocols: (i) conventional 10 min at 25 °C followed by 10 min at 50 °C, or (ii) 10 min at 50 °C, or (iii) 5 min at 50 °C, or (iv) 5 min at 30 °C. Following each transcription protocol, the RTase was inactivated by a 5 min incubation at 85 °C (SS4) or 95 °C (US2). The tubes were cooled and kept at 0 °C for five minutes inside the heating block. The tubes were removed, spun for 1 min at 3K RPM, placed on ice and the cDNA diluted with 10 µL of water. Contents were mixed by pipetting and 1 µL aliquots were used for further qPCR analysis.

Alternatively, for reactions carried out without a dedicated RT step, reagents, tubes and RNA were kept at room temperature, which fluctuated between 19 and 22 °C. These reactions were set up on the bench. Since the time taken to dispense reagents, aliquot the reactants into tubes and spin them depended on the number of samples being processed, we standardised the time to five minutes for all samples. In some instances, 5 µL aliquots of complete reaction mix and RNA were kept on ice for five, ten or fifteen minutes. The tubes were placed into the G-Storm thermal block pre-heated to either 85 °C (SS4) or 95 °C (US2) for five minutes, cooled and kept at 0 °C for five minutes inside the heating block. Tubes were spun for 1 min at 3K RPM and placed on ice. The cDNA was diluted with 10 µL of water, contents were mixed by pipetting and 1 µL was used for each reaction for qPCR analysis.

For RT gradients, reactions were set up on ice using pre-cooled reagents with 5 µL volumes dispensed into rows A to H in 96-well qPCR plates kept on ice. Plates were sealed at room temperature on a PX1 plate sealer and immediately spun for 15 s at 2K RPM at 0 °C in a benchtop centrifuge. Plates were transferred into the BioRad CFX qPCR instrument programmed to run for 5 min at eight temperatures in a gradient ranging from 30 °C (row H, minimum temperature possible) to 50 °C (row A), followed by 5 min at 85 °C (SS4) or 95 °C (US2). The plates were removed from the instrument and spun for 5 min at 2K RPM at 0 °C. The seals were pierced, and the cDNA from each well was recovered. A volume of 10 µL of water was added to each well and the plates were spun again for 1 min. The liquid was recovered and added to the cognate cDNA. A volume of 1 µL of the combined cDNA was used for further qPCR analysis.

qPCR reactions were carried out for 40 cycles with 1 s denaturation at 95 °C and 1 s polymerisation at 60 °C. When SYBR Green was used to detect PCR amplicons instead of probes, an additional melt curve was programmed into the run.

### 4.5. 1-Step RT-qPCR Reactions

Legacy RT-qPCR reactions incorporated defined RT steps at 50 or 55 °C carried out for variable amounts of time, followed by a 30 s to 1 min 95 °C *Taq* activation step and 40 cycles of 1 s denaturation at 95 °C and polymerisation at 60 °C, respectively. Reactions were set up on ice using pre-cooled reagents and qPCR plates, into which 5 µL volumes of master mix and RNA were dispensed. For CFX runs, 96-well plates were sealed at room temperature on a PX1 plate sealer, spun for 15 s at 2K RPM at 0 °C in a benchtop centrifuge and transferred to the qPCR instrument. For Eco runs, an adhesive seal was applied to the 48-well plates on ice, spun for 5 s at 2K RPM at 0 °C in a benchtop centrifuge and transferred to the qPCR instrument. For legacy runs carried out on the BMS Mic instrument, tubes were sealed and immediately placed inside the instrument. The RT-qPCR was carried out using a 1 min RT step at 55 °C followed by a 30 s 95 °C activation step and 40 cycles of 95 °C denaturation for 5 s and 60 °C polymerisation for 10 s. The details of the various RT reaction conditions tested are described in the following text and legends. 

When conditions needed to be modified due to the inability of the master mix to utilise fast reactions conditions, this is indicated in the applicable text and legends.

For RT-qPCR assays pre-run with gradients for the RT step, reactions were set up on ice using pre-cooled reagents and 96-well plates. The 5 µL volumes were dispensed into rows A to H of the qPCR plate, the plates sealed at room temperature on a PX1 plate sealer and spun for 15 s at 2K RPM, 0 °C in a benchtop centrifuge. Plates were transferred to the BioRad CFX qPCR instrument programmed to run for 5 min at eight temperatures from the instrument minimum 30 °C (row H) to 50 °C (row A), followed by 2 min activation of the *Taq* at 95 °C and 40 cycles of 1 s denaturation at 95 °C and polymerisation at 60 °C, respectively. Any modifications are described in the applicable text or legends.

Reactions carried out without a dedicated RT step used reagents, tubes, qPCR plates and RNA kept at room temperature. However, where the aim was to determine RT efficiency at 0 °C, reactions were set up on ice with pre-cooled reagents, as described above. As with the 2-step protocols, the process of dispensing and setting up the reactions was standardised to five minutes for all samples. Plates were handled as before with the 2-step reactions, except that samples were immediately amplified without an RT step using a standard qPCR protocol of 40 cycles, generally with 1 s denaturation and 1 s polymerisation steps. If this needed to be modified due to the inability of the master mix to utilise fast reaction conditions, this is indicated in the applicable text and legends.

## Figures and Tables

**Figure 1 ijms-23-01303-f001:**
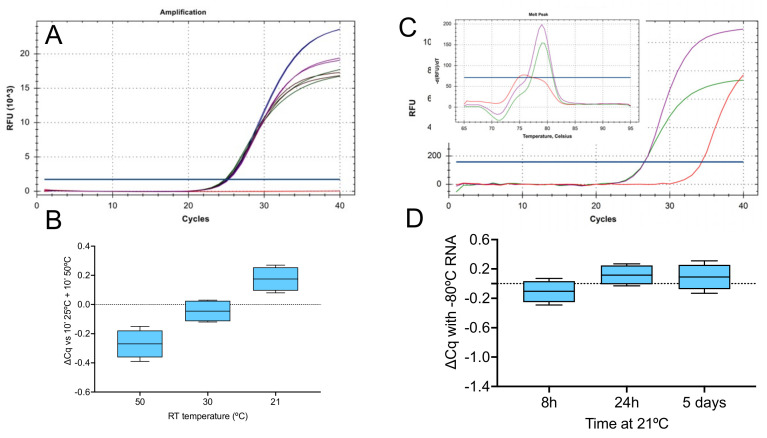
RT-qPCR of RNA reverse transcribed by SS4 and targeting the viral E-gene (assay E1). (**A**) Amplification plots and Cqs from duplicate qPCR reactions detected by a hydrolysis probe: conventional 10 min at 25 °C and 50 °C (blue), 5 min at 50 °C (green), 30 °C (brown) or 21 °C (purple). (**B**) ∆Cq for the three modified conditions relative to the conventional protocol. (**C**) Amplification plots and melt curves (inset) from single qPCR reactions: 5 min at 50 °C (green) or 21 °C (purple). The NTC is shown in red. (**D**) Cq values of RNA samples left on the bench for the indicated times relative to control sample kept frozen at −80 °C. Box and whiskers plots show the ∆Cqs for the independent RT-qPCR reaction, with the whiskers delineating the minimum and maximum values.

**Figure 2 ijms-23-01303-f002:**
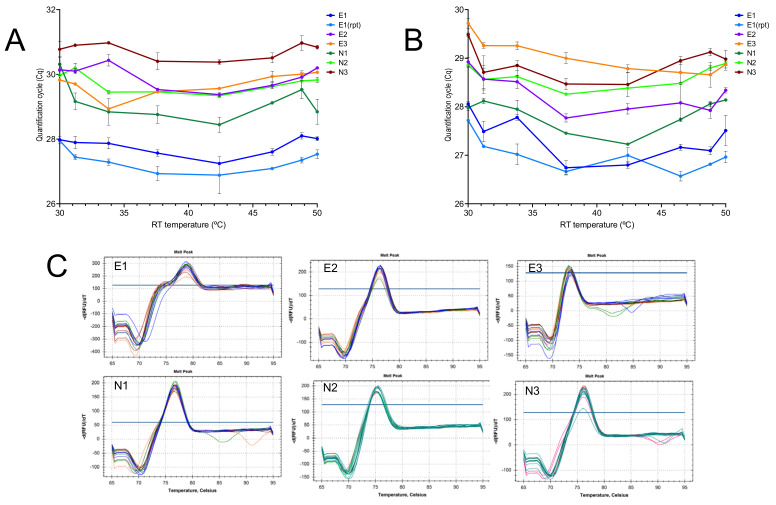
Amplification with assays E1, E2, E3 and N1, N2 and N3 of RNA reverse transcribed by SS4 on an RT gradient. (**A**) Plot of Cqs vs. RT temperature gradient, with PCR amplicons detected by hydrolysis probes. (**B**) Plot of repeat reaction detected using SYBR Green. (**C**) Melt curves for the six assays.

**Figure 3 ijms-23-01303-f003:**
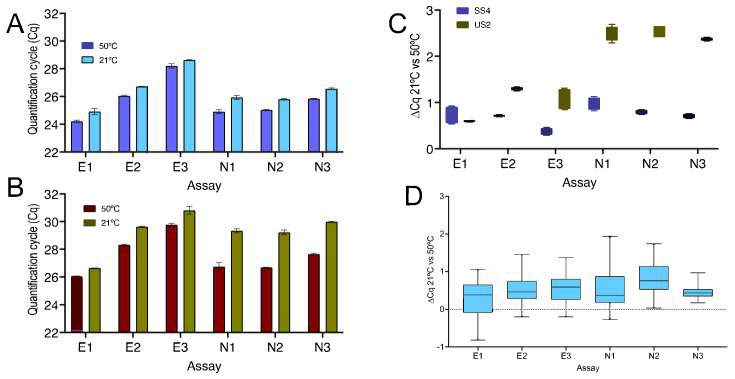
Comparison of RNA reverse transcribed by SS4 and US2, with PCR amplicons from assays E1, E2, E3, N1, N2 and N3 detected by hydrolysis probes. (**A**) SS4: Cqs from duplicate qPCR reactions; 5 min at 50 °C (dark blue) or 21 °C (light blue). (**B**) US2: Cqs from duplicate qPCR reactions; 5 min at 50 °C (dark brown) or 21 °C (olive). (**C**) Box and whiskers plot comparing the ∆Cq (21 °C vs. 50 °C) for these assays: SS4 (blue) and US2 (olive). (**D**) ∆Cqs of RT-qPCR reactions carried out following three and nine independent 5 min RT reactions at 50 and 21 °C, respectively. All box and whiskers plots show the ∆Cqs for the independent RT-qPCR reaction, with the whiskers delineating the minimum and maximum values.

**Figure 4 ijms-23-01303-f004:**
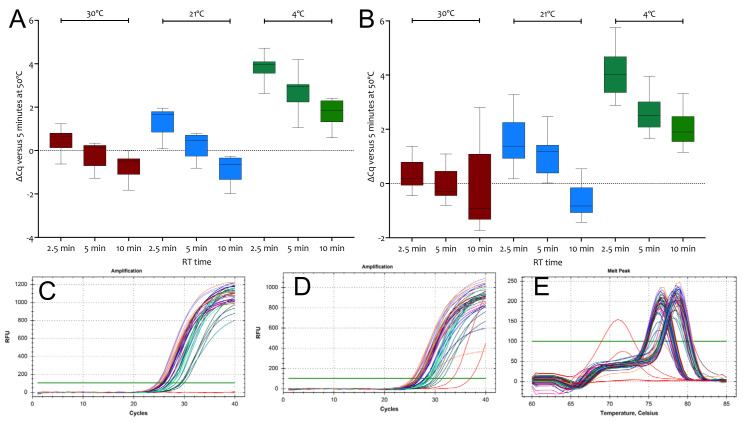
∆Cq values recorded for RT reactions carried out at for 2.5, 5 or 10 min at 30 °C (brown), 21 °C (blue) or on ice (green) relative to 5 min RT reactions at 50 °C. (**A**) Cq values recorded for assay E1. (**B**) Cq values recorded for assay N1. (**C**) Amplification plots for E1. The NTCs (red) did not record Cqs. (**D**) Amplification plots for N1. Both NTCs recorded Cqs (red). (**E**) Melt curves for E1 and N1. The two positive NTCs display melt curves different from those obtained for assay N1. All box and whiskers plots show the ∆Cqs for the independent RT-qPCR reaction, with the whiskers delineating the minimum and maximum values.

**Figure 5 ijms-23-01303-f005:**
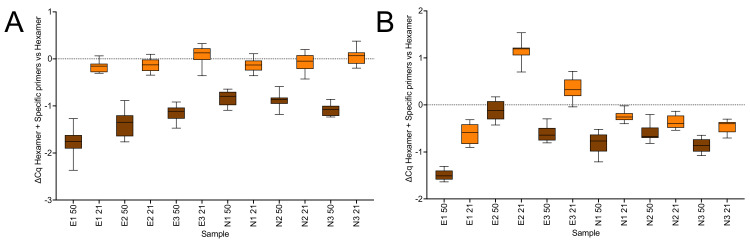
Amplification of RNA reverse transcribed with random hexamers only or with additional specific reverse primers targeting the E (E1–3) and N (N1–3) genes, respectively, and detected using hydrolysis probes. (**A**) SS4: ∆Cqs for RT reactions carried out with and without specific reverse primers at 50 °C (brown) and at 21 °C (orange). (**B**) US2: ∆Cq for the reactions carried out with and without specific reverse primers at 50 °C (brown) and at 21 °C (orange). All box and whiskers plots show the ∆Cqs for the independent RT-qPCR reaction, with the whiskers delineating the minimum and maximum values.

**Figure 6 ijms-23-01303-f006:**
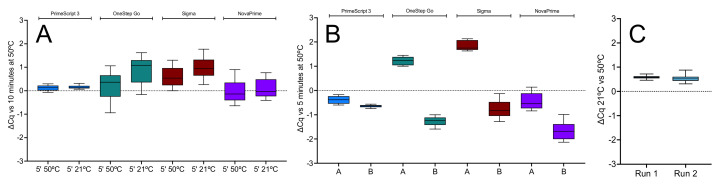
The 1-step RT-qPCR assays targeting E1 carried out with PrimeScript 3 (blue), OneStep Go (green), Sigma (brown) and NovaPrime (purple) master mixes. (**A**) ∆Cqs of 5 min RT reactions carried out at 50 or at 21 °C against 10 min RT reactions. (**B**) ∆Cqs from two replicate 5 min RT reactions (**A**,**B**) carried out at 21 °C relative to a 5 min RT reaction carried out at 50 °C. (**C**) ∆Cqs recorded for two runs of ten RT replicates of the E1 assay on the BMS Mic instrument. The scales of the y-axes were adjusted to a common scale to simplify the comparison between the runs and instruments. All box and whiskers plots show the ∆Cqs for the independent RT-qPCR reaction, with the whiskers delineating the minimum and maximum values.

**Figure 7 ijms-23-01303-f007:**
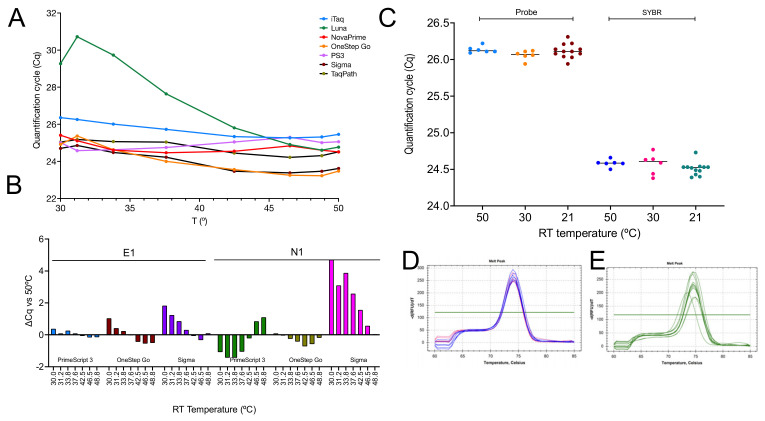
RT temperature gradients demonstrating low-temperature transcription by RTases. (**A**) Plot of Cqs vs. RT temperature gradient for E1 amplified by seven 1-step master mixes. (**B**) ∆Cq values for the E1 and N1 assays recorded by three 1-step master mixes. (**C**) Cqs demonstrating the reproducibility of RT results using PrimeScript 3. E1 PCR amplicons were detected with a FAM-based probe (light blue, orange and brown) or SYBR Green (dark blue, pink and green). There are six replicates at 50 and 30 °C and twelve replicates at 21 °C. (**D**) Melt curves for PCR amplicons obtained following RT at 50 °C (dark blue) and 30 °C (pink). (**E**) Melt curves for PCR amplicons obtained following RT at 21 °C.

**Figure 8 ijms-23-01303-f008:**
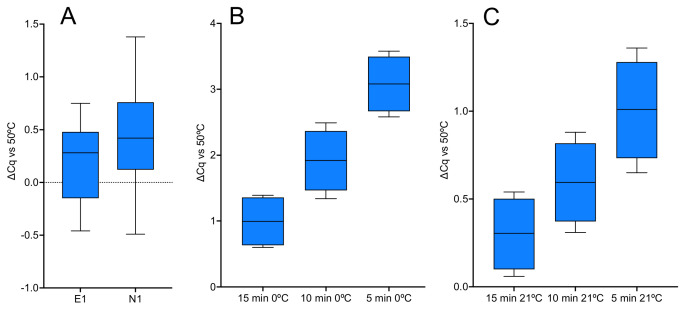
RT activity at 0 or 21 °C using PrimeScript 3. All box and whiskers plots show ∆Cqs relative to the 5 min at 50 °C control. (**A**) ∆Cqs for E1 and N1 assays left on ice for 15 min. (**B**) ∆Cqs from a time course of E1 reactions left on ice for 15, 10 or 5 min. (**C**) ∆Cqs from a time course of E1 reactions left at 21 °C for 15, 10 or 5 min.

**Figure 9 ijms-23-01303-f009:**
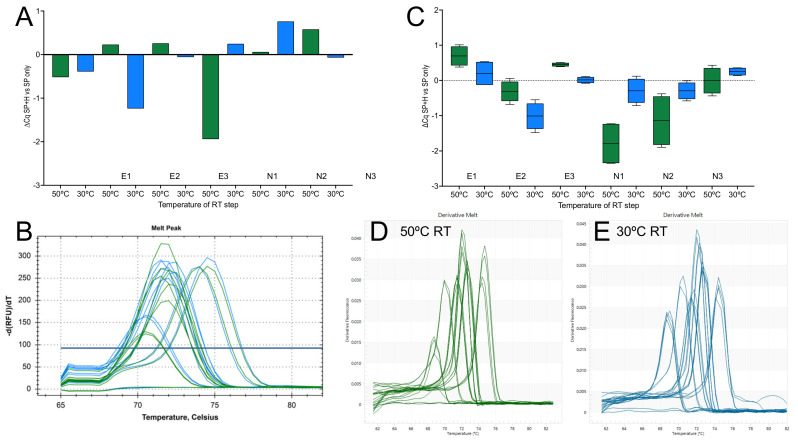
Effect of random primers on 1-step RT-qPCR assays. (**A**) ∆Cq values (100 nM random and specific primers versus specific primers only). RT reactions were carried out on the CFX at 50 °C (green) or at 30 °C (blue). (**B**) Melt curves from the above assays. (**C**) ∆Cq values (100 nM random and specific primers versus specific primers only). RT reactions werecarried out at 50 °C (green) or at 21 °C (blue) on the Eco. (**D**) Melt curves from the above 50 °C assays. (**E**) Melt curves from the above 30 °C assays.

**Table 1 ijms-23-01303-t001:** Primer, probe and amplicon sequences and characteristics for the six assays targeting SARS-CoV-2 RNA. Capital letters within the probe sequences mark LNA substitutions.

E-gene 26,305–26,429 (NC_045512.2)						
Assay	Primers (5′-3′)	Tm (°C)	R primer position	FAM Probe (5′-3′)	Tm (°C)	Amplicon (bp)
E1-F	CTTGCTTTCGTGGTATTCTTG	63.5		actAgcCatCctTactgcg	71.1	69
E1-R	GCAGTACGCACACAATCG	65.2	26,356–26,373			
E2-F	CTTCGATTGTGTGCGTAC	62.0		tgaGtcTtgTaaAaccttc	64.9	77
E2-R	ACACGAGAGTAAACGTAAAAAG	63.0	26,408–26,429			
E3-F	TGCTGCAATATTGTTAAC	57.0				56
E3-R	CGAGAGTAAACGTAAAAAG	57.0	26,408–26,426			
**Nsp10 13,025–13,094 (NC_045512.2)**						
	Primers (5′-3′)	Tm (°C)	R primer position	FAM Probe (5′-3′)	Tm (°C)	Amplicon (bp)
N1-F	CTGGTAATGCAACAGAAG	57.1		ctgCcaAttCaaCtgta	65.2	69
N1-R	CAGCATCTACAGCAAAAG	57.5	13,077–13,094			
N2-F	CTGGTAATGCAACAGAAG	57.1				59
N2-R	AGCAAAAGCACAGAAAG	56.8	13,068–13,084			
N3-F	GCTGGTAATGCAACAGAAGTG	63.9				61
N3-R	CAGCAAAAGCACAGAAAGATAAT	63.7	13,063–13,085			

## Data Availability

All raw data reported in this study are included in the Appendix A.

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
