# Peer review of "RT-qPCR Detection of SARS-CoV-2: No Need for a Dedicated Reverse Transcription Step"

_ijms, 2022, doi:10.3390/ijms23031303_

Round 1

Reviewer 1 Report

Indeed, it is unusual but I find this paper very clear and well structured, both in terms of the language as well as in terms of the scientific experiments and data analysis.

The authors investigated the performance of doing the reverse transcription step of an RT-qPCR reaction at room temperature instead of at the usual high temperature. They show that this does not impact the quantitative estimate of the qPCR. As the authors state, this is a paradigm shift and will help design even faster RT-qPCR diagnostics.

This is a clearly written paper on a topic that will be of interest to the users. The design of the experiments, presentation of the data and discussion of the results are sound. 

I have no further comments on this paper

Reviewer 2 Report

The manuscript submitted by Bustin et al. and entitled “RT-qPCR detection of SARS-CoV-2: no need for a dedicated reverse transcript step” describes a study to develop methods to reduce the time needed for RT step prior to testing of a qPCR assay.  In this work the authors look at 2-step and 1 step RT-PCR methods.  By using short amplicons and specific RTase enzymes they can get reliable amplification without the need for long RT steps at high temps.  These data are demonstrated by testing temperature gradients compared to standard RT methods.  They compare testing using multiple enzymes and multiple targets (E-gene and N-gene).  Changes in Cqs were minimum at room temperature testing, but there was more than 2 Cq changes when tested on ice in most cases.  Although this works, the loss of sensitivity may be an issue for clinical testing for 4C.  Overall, the manuscript is well written and easy to follow.  Similar data was found in Rejali et al in 2020, but this work was performed only with Zika virus as a target, but this manuscript adds in the target of SARS-CoV-2 and looks specifically at lower temperatures.  There are a few comments that need to be addressed prior to publication.

Major Comments

RNA source: You reference one of your previous publications and that they were obtained from Mid and South Essex NHS foundation.  However, it would be helpful to understand a bit more of these samples to evaluate the study.  Are these patient samples, control material, if only control material, is there risk of amplification from patient samples that would have other organisms present?  Also would a different extraction method cause issues with amplification at 21-30C?

Based on the Cq values in your figures the RNA testing appears to be with moderately positive samples.  As changes in protocols most often effect at the extremes it would be essential to know how well these methods work closer to the limit of detection of the assay.  Would you get efficient RT and amplification if viral load was low in >30 Cq specimens.  I would also add high viral load samples as well in the event a high-dose hook effect is created.

Discussion: The work done in this is all on use of short amplicons; however, there are limitations to using short amplicons such as effects of mutations that could reduce amplification.  The authors should go into discussion about limitations of short amplicons.

Minor comments

Figure 3 – Although its in the caption, it would be helpful to add figure legends to quickly differentiate the different colors in the graph.

Discussion: In general, it is a bit long and redundant of the results section.  I would suggest removing some of the results.
